# Organizational Food Environments: Advancing Their Conceptual Model

**DOI:** 10.3390/foods11070993

**Published:** 2022-03-29

**Authors:** Inês Rugani Ribeiro de Castro, Daniela Silva Canella

**Affiliations:** 1Department of Social Nutrition, Institute of Nutrition, Rio de Janeiro State University, Rio de Janeiro 20550-900, Brazil; inesrrc@uol.com.br; 2Department of Applied Nutrition, Institute of Nutrition, Rio de Janeiro State University, Rio de Janeiro 20550-900, Brazil

**Keywords:** food environment, workplace, university, school, framework

## Abstract

Understanding the complexity of the elements that constitute organizational food environments and their operating dynamics is essential to improving their healthiness. This study developed a conceptual model of organizational food environments. For this purpose, a comprehensive literature review was conducted, a first version of the conceptual model was prepared, a panel of experts was consulted, the model was improved, a second panel of experts was consulted, and the model was finalized. The model consists of four components (the institutional level, internal level of eating spaces, surroundings, and the decisional level) and 10 dimensions related to the institutional level and internal level of eating spaces (the availability, accessibility, affordability, quality, food and nutrition information, and promotion of foods, beverages, and culinary preparations and the availability, acceptability, convenience, ambience, and infrastructure of the eating space), as expressed in a graphical scheme. The conceptual model presented here offers innovative elements which contribute to understanding of the organizational food environment. It can guide the development of both assessment studies of food environments and interventions for their improvement.

## 1. Introduction

At the end of the 1990s, as obesity grew more prevalent across the world, discussion on the insufficiency of individual approaches to cope with obesity was encouraged. One focus was that environmental interventions should be prioritized [1]. Since then, researchers have made efforts to understand the role of food environments and to define, typify, and characterize them to facilitate their improvement [2,3,4,5].

A food environment can be defined as the collective physical, economic, policy and sociocultural surroundings, opportunities and conditions that influence people’s food and beverage choices and nutritional status [1,2,3,4,5,6,7]. These environments interact with supply chains and individuals, influencing and being influenced by them [7]. They are the object of public policies, national plans, and international recommendations to promote adequate and healthy eating [7,8,9,10,11].

One type of food environment present in some theoretical models is the organizational food environment [2,3]. The definition adopted in the present study is the one proposed by Gálvez Espinoza et al. [3]: “A place where food is sold or supplied to workers, students, or other members working in institutions and organizations. It includes schools, universities, companies, public services, hospitals, prisons, and civil society associations and their respective food centers (cafeterias, kiosks, and food vending machines)”.

Models that include organizational food environments in their structure make little use of their dimensions [2,3]. In studies measuring such dimensions, the presence of eating spaces and the availability and affordability of food have received considerable attention [12,13]. A broader understanding of the complexity of the elements that constitute food environments and their dynamics of operation is essential to making them more health-promoting [3,14]. In this sense, the present article describes the development of a conceptual model of organizational food environments and details its components to help understand the elements that make up an organizational food environment. Our purpose is to contribute to overcoming the approach to food environments that has been restricted to the availability of food in organizations’ eating spaces.

## 2. Methodological Approach

The development of the proposed conceptual model for the organizational food environment included five stages: a literature review, insight from our previous experience assessing organizational environments, evaluations of the model by experts, dialog with other actors in scientific disciplines and conferences, and the development of a final proposal for the model.

The extensive literature review identified theoretical models that considered organizational food environments, as well as dimensions of food environments in general that could be included in the proposed model [1,2,12,13,15,16,17]. It also helped us identify elements to be included from the previous experience of the researchers in conducting studies evaluating food environments in the workplace [18], universities [19], hospitals [20] and schools [21].

Based on these findings, the first version of the model was conceived. This version contemplated the institutional, establishment, and decisional levels and the dimensions of availability, accessibility, affordability, convenience [16], nutritional information, advertising [2], ambience [22], infrastructure for food [23], and access to water [23,24].

Aiming to solicit their opinions on the first version of the conceptual model, we held a panel in August 2018 with 11 experts with extensive experience in the topics at hand: three from the area of collective feeding, six from the field of epidemiology and public health, and two managers of schools and public hospitals. The literature recommends the participation of five to 20 professionals in this stage [25]. The experts previously received the study presentation material, containing the theoretical framework to support the discussion, as well as the first version of the graphical scheme of the conceptual model and the definitions of its components and dimensions. The discussion was conducted through a face-to-face workshop in which group and plenary discussions were held. The suggestions were later analyzed, and relevant elements were incorporated into the conceptual model.

We also made use of interactions with experts in food environments during scientific events, such as the II Latin American Seminar on Food Environments and Health, held in June 2019 in Rio de Janeiro, and discussions in the classroom in a class taught since 2017 to graduate students on the topic of food environments.

A second evaluation by experts was performed in August 2021 to evaluate an improved version of the model. In a virtual consultation, the experts gave their opinions about the new graphical scheme, the components, and the dimensions proposed for the model. For this, the experts answered 13 questions on the relevance, comprehensiveness, and comprehensibility of each of the elements of the model on a four-point Likert scale (strongly agree, partially agree, partially disagree, strongly disagree). After each question, the experts could add their free-response observations. Nine experts in the field of nutrition participated. They had experience in the subject of food environments and expertise in epidemiology and collective health or in collective feeding. The suggestions and notes given by the experts were discussed among the authors of the study. They were then incorporated into the conceptual model when it converged with its proposal and contributed to its improvement.

During the elaboration of the final version of the conceptual model presented here, the following procedures were carried out. First, the nomenclature and definition of its components and dimensions were revised, incorporating input from other references [5,26,27,28,29,30,31]. Next, the surroundings component, which encompasses physical and virtual contexts, was included. Then, the quality dimension was included, followed by the informal market at the institutional level. After the above activities (literature review, consultation with experts, discussion with other actors in different spaces), a final version of the conceptual model was proposed. It has four components (the institutional level, internal level of eating spaces, surroundings, decisional level), 10 dimensions referring to the institutional level and internal level of the eating spaces, and a graphical scheme.

## 3. Results

The final model is shown in Figure 1. Definitions and examples of each of its four components are presented below.

### 3.1. Components of the Conceptual Model

#### 3.1.1. Institutional Level

The institutional level encompasses the elements of the physical environment existing in the organization that influence food choices and practices, the set of eating spaces (which include commercial food services, noncommercial food services, vending machines, and mini-kitchens) made available, managed, or contracted by it, and the informal market that occurs under its auspices. Examples include the existence of commercial or noncommercial food services, vending machines, and/or mini-kitchens; the supply of water and coffee; and the sale of food products by members of the organization (workers, students, etc.), street vendors, and informal vendors. Mini-kitchens are facilities restricted to the internal members of the organization to let them eat food taken from home, with a microwave and/or refrigerator, tables, and chairs. Commercial food services are food services whose main purpose is preparing and selling food and beverages [32]. Noncommercial food services are those where food and beverages are not the primary focus of a business but rather where food and beverages are present to support or supplement a specific group (e.g., employees of an organization) [32]. The informal market is often not regulated through formal governance structures [5,33] but rather is carried out by mobile vendors or members of the organization.

This component of the model dialogs with the concept of community nutrition environment proposed by Glanz et al. (2005) [2]: “the distribution of food sources, that is, the number, type, and location and accessibility of food outlets”. Although the authors are referring to territories, their definition can be useful in assessing the food environment of organizations, especially those that are more complex, such as universities, hospitals, and companies located in large buildings.

#### 3.1.2. Internal Level of the Eating Spaces

The internal level encompasses the elements of the food environment inside each eating space (including commercial and noncommercial services, vending machines, and mini-kitchens). Examples include available food, forms of payment, opening hours, prices, food and nutrition information, promotion, characteristics of the infrastructure and adequacy of the installed capacity for storage, preparation (when appropriate), heating (when appropriate), and having meals. This component of the model is related to the concept of consumer nutrition environment proposed by Glanz et al. (2005) [2]: “what consumers encounter within and around a retail food outlet (i.e., store or restaurant), and most of these characteristics will also apply to food sources in organizational environments”.

#### 3.1.3. Surroundings

Surroundings refer to the physical and “virtual” contexts related to food that are available to people who attend a particular organizational environment and that are not interfered with by the management of this organization. The physical context includes the establishments that sell food, beverages, and culinary preparations, as well as the informal marketing of these products in the area adjacent to the organization. It also includes public spaces that may favor (e.g., a park with trees and tables) or not (e.g., inhospitable territory or with a high crime rate) eating outside the organization. The “virtual” context refers to the formal and informal market of food, beverages, and culinary preparations that materialize within the organization by demand of the people who attend or work at the organization. The scope of this context will depend on the logistical capacity of the delivery services.

On a two-way street, the surroundings influence and are influenced by how the organization structures the food options for its members. For example, an organization will seek different arrangements to offer food, meals, and eating spaces to its members depending on the choices of food establishments that exist in its surroundings. On the other hand, depending on the number and profile of members of an organization and its operating dynamics, the surroundings can change to meet its food demands.

#### 3.1.4. Decisional Level

The decisional level refers to the governance of the organization’s food environment, which occurs in two spheres: external and internal to the organization. It concerns power relations (power to, power over) [34] and decision-making processes about this environment.

The external sphere encompasses national and subnational policies, laws, and regulations that regulate the dynamics of the functioning of organizations [3]. Examples include health legislation, legal frameworks that guide the work dynamics, and the ordering of the physical structure of eating spaces and other spaces of the organization.

The internal sphere encompasses instances and arenas, agents, and processes involved in decision-making within the organization [35] that interfere with the food environment. The interaction and reciprocal effect between these three elements determine the conformation of the food environment. Examples include instances and arenas, including instances that are responsible for the different elements of the food environment of the organization (sectors included in the organization chart) and both formal (councils, collegiate bodies, etc.) and informal arenas (collective of members of the institution, etc.) for decision-making; agents, including managers and representatives of the various segments that make up the organization; centralized, decentralized, and shared processes; norms/rules; and institutional culture. Notably, in this component of the conceptual model, we are not referring to decision-making at the individual level in the sense of the individual’s food choices. Our focus is the decision-making processes that shape the food environment, that is, what determines it.

### 3.2. Dimensions of the Food Environment at the Institutional Level and Internal Level of Eating Spaces

To better understand the complexity of the institutional level and the internal level of eating spaces, we present their dimensions. They are decisive for the food choices of those who live and work in these environments. These dimensions include the availability, accessibility, affordability, quality, food and nutrition information, and promotion of foods, beverages, and culinary preparations and the availability, acceptability, convenience, ambience, and infrastructure of the eating spaces. Table 1 summarizes the definitions and examples of these dimensions and indicates which component(s) of the model each one of them refers to.

Many of these dimensions would also apply to the “surroundings” component of the proposed conceptual model. However, detailing this component is beyond the scope of this article, which focuses on the organization itself and the elements intrinsic to it. Regarding the decisional level, the purpose of the present study is to present this component of the model in a pioneering manner. A deeper understanding of its complexity through the proposition of its dimensions will be the object of further study.

## 4. Discussion

The conceptual model presented here offers the following contributions to current knowledge on the subject:(a)It systematizes tangible and intangible elements that shape the organizational food environment and ultimately influence the food choices and practices of individuals exposed to it.(b)It explains the complexity of this food environment by structuring the environment into four components: the institutional level, internal level of eating spaces, decisional level, and surroundings.(c)It introduces a new element in the debate on the subject: the governance of organizational food environments, as expressed in the decisional level component.(d)It explains the informal market as an element of this environment (both in the surroundings and at the institutional level), which is often a very relevant element in some realities [5].(e)It includes the “virtual” surroundings, which have received increasing attention in the debate on food environments [28].(f)It explains the reciprocal influence between the surroundings and the organization.(g)It recognizes that, in addition to the food offered (commercialized or not), the infrastructure that allows members of the organization to take food from home and eat comfortably and safely also plays an important role in shaping this environment.(h)It includes the quality dimension of the foods, beverages, and culinary preparations offered and adopts a holistic approach to this polysemic concept. It incorporates elements related to culture and sustainability (forms of production and industrial processing), in addition to nutritional composition, sensory characteristics, and health safety [5,26,27,31], encompassing aspects related to the production, extent, and purpose of industrial processing [29], cultural references, nutritional attributes, and health safety (e.g., microbiological safety), among others.

The incorporation of the governance of the organizational food environment represents an important innovation in efforts to understand this environment. The improvement of organizational food environments has been identified as a relevant strategy for the promotion of healthy eating, health, and equity [36,37,38]. However, the debate on power relations and the correlation of forces between the agents involved in decision-making processes in this environment, as well as what the decision-making bodies are and how democratic and transparent they are, is neglected.

For example, in a process of assessing the food environment of a given organization, besides identifying the places where food is offered, the factors determining what is sold inside are worth investigating. Who prepares the terms of reference (food procurement contracts) that will guide the food service contract offered? Which department of an organization decides on the time dedicated to meals, how its members will have access to water, and the extent to which the institution’s spaces will be structured to ensure the necessary infrastructure for those who want to eat food brought from home? If this component of the organizational food environment is not visualized, a risk of exhaustively describing how the food environment is configured without knowing the factors and forces influencing that environment arises. The improvement of these environments presupposes the political will to change and the decision-making capacity for this change.

As a limitation of this study, the literature review did not follow internationally recommended protocols, such as PRISMA-ScR. However, throughout the process of elaborating the proposed model, the literature was reviewed and updated through regular surveys of bibliographic databases, journals that have published studies on food environments, publications by organizations linked to the United Nations, and theses and dissertations conducted in recent years. These searches were complemented by the sharing of publications in conferences on food environments and in classes offered in graduate programs in Brazil.

The proposed model provides the conceptual basis for measuring food environments. It has already been used by José et al. (2021) [39] for the development of an instrument that measures hospital food environments in Brazil. Notably, however, measuring organizational food environments entails many challenges, including the development of mechanisms for assessing the decisional level, the application of indicators of the healthiness of the organizational environment [21,40], and the improvement of procedures for assessing the informal market and its surroundings. Regarding the latter, one challenge is to delimit what will be investigated considering the specificities that are of interest to the organizational food environment. For example, establishments that sell ready-to-eat foods near sites where its members work, which they can then buy and consume on their lunch/snack break. Another challenge is to improve the procedures for measuring the virtual context. Notably, the components of the model are dynamic, and their measurement should consider this dynamism.

The proposed model also provides support for the design of interventions and public policies, as it contributes to the understanding of the constituent elements of the organizational food environment. From this perspective, this study adds to the efforts already undertaken by Swinburn et al. (1999) [1] and Kanter et al. (2015) [41], for example, in the sense of offering conceptual models to subsidize public policies involving food environments.

## 5. Conclusions

By conceiving the organizational food environment in four components (the institutional level, internal level of eating spaces, decisional level, and surroundings), the proposed conceptual model contributes to overcoming the current approach, which has been restricted to food availability in organizations’ eating spaces. This conceptual model can guide the development of interventions to improve this environment and studies to measure it. Accordingly, future efforts are needed to develop and improve tools and indicators for an accurate assessment of the proposed components and dimensions.

## Figures and Tables

**Figure 1 foods-11-00993-f001:**
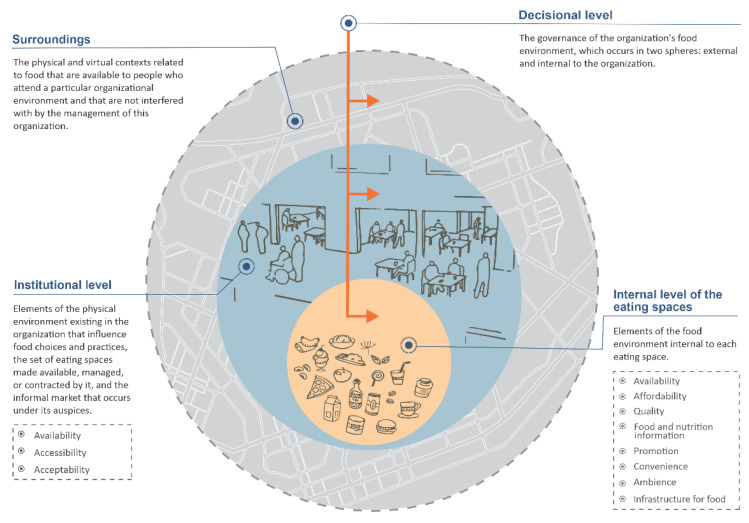
Conceptual model of organizational food environments.

**Table 1 foods-11-00993-t001:** Definitions and examples of the dimensions of the food environment at the institutional level and internal level of food spaces.

Dimensions	Definition and Examples
Availability	Presence of eating spaces and other forms of food sale/supply within the organization. *(Institutional level)* Examples: commercial and noncommercial food services, vending machines, mini-kitchens, informal market, and delivery. Availability of water for members of the organization. *(Institutional level)* Examples: water fountains in the corridors, filters in mini-kitchens, water distributed in bottles. Presence of foods, beverages, and culinary preparations within the eating spaces. *(Internal level of eating spaces)* Examples: fresh or minimally processed, processed, and ultra-processed foods, beverages, and culinary preparations based on one or more of these groups.
Accessibility	Ease or difficulty of reaching the eating spaces and/or water supply points. *(Institutional level)*
Affordability	Food prices relative to the purchasing power of individuals. *(Internal level of eating spaces)*
Quality	Includes elements such as the production process (agroecological or not; pesticide use, genetic modification technology use, among others), the extent and purpose of industrial processing, cultural reference (tradition, knowledge of origin), nutritional composition, health security (microbes, other contaminants such as heavy metals, pesticides), integrity, freshness, and other sensory attributes of foods, beverages, and culinary preparations. *(Internal level of eating spaces)*
Food and Nutrition Information	Refers to statements of energy and nutrient value on the label of packaged foods, unpackaged foods, and on the menu, as well as information on the foodstuffs used in culinary preparations and their origin. *(Internal level of eating spaces)*
Promotion	Refers to marketing communication strategies, as well as other communication and educational strategies to promote foods, beverages, and culinary preparations. It also includes strategies related to food prices, such as combos (food + accompaniment (drink or dessert) at a more attractive price than if purchased separately), larger portions of the same product at promotional prices. *(Internal level of eating spaces)* Examples: displays, posters, brochures, food replicas with messages to stimulate their consumption, 600 g portion of French fries with a price less than twice that of the 300 g portion.
Acceptability	Refers to the attitudes of people about their local food environment and whether the supply of products meets their personal standards. *(Institutional level)*
Convenience	Existence of elements facilitating the acquisition of foods, beverages, and culinary preparations, such as opening hours, payment method, and availability of delivery services, that meet the needs of the members of the organization. *(Internal level of eating spaces)*
Ambience	Comprises components that involve the participants, influencing physiology, motivation, mood, behavior, cognition, and social interaction, such as thermal comfort, lighting, noise level, and hygiene of the physical space. *(Internal level of eating spaces)*
Infrastructure for food	Refers to the internal infrastructure for meals in the eating spaces. *(Internal level of eating spaces)* Examples: existence of equipment to store and heat food, beverages and culinary preparations, and furniture and utensils for meals.

Component of the model to which the dimension refers is presented in italic.

## Data Availability

Not applicable.

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
