# Peer review of "Organizational Food Environments: Advancing Their Conceptual Model"

_foods, 2022, doi:10.3390/foods11070993_

Round 1

Reviewer 1 Report

The article tackles an interesting research question. Unfortunately it is written in a way that it is very hard to read. Please revise the English, write shorter sentences and use less listings. Sentences with 11 words are easy to read, 21 words are fairly difficult, and everything above is very hard to follow. The first sentence of the methodology has 49 words, this is impossible to read properly! also the first sentence of subchapter 3.2 has 102 words!!! Check also your citation style in text for example Glanz et al. [2] (Line 129) or Glanz et al. (2005) (Line 142)?

Methodology:

Line 84 to 94: In the second consultation round: how was decided if comments of experts were relevant or not?

Line 95 to 99: Please write a coherent text or provide a flowchart for clarification.

Results:

Figure 1: I suggest to use bulletpoint over sentences. By first glance, the figure is not understandable. The dimensions are not well illustrated.

Line 112 to 119 and line 136 to 138: This is self-plagiatism of the figure. Avoid that by changing the sentence in the figure to bulletpoints or just a few important words.

Chapter 3.2: this chapter needs to be rewriten. The dimensions should be well described in the text and only examples left in the table (Table 1). The way it is right now, is very confusing.

Discussion:

The discussion is a huge list of of contributions and examples. Please write a coherent text or put the lists in another format. I understand that you have lists, but this looks like you put a powerpoint presentation in a manuscript.

Conclusion:

The conclusion is too short.

Reviewer 2 Report

The article "Organizational food environments: Advancing their conceptual model" can be accepted for publication.
 It is clear and well organized; it just needs to be subject to some minor revision:

- In the introduction, it would be better to add some bibliography regarding the concept of "food environment". 

- Otherwise, it is clear and well articulated.

- In the conclusion, it is clear that this conceptual model represents an innovation. It is asked to articulate the conclusions, making the text more substantial and inserting a sentence about future prospects.

Round 2

Reviewer 1 Report

The article is still hard to read and sentences are too long. Please revise the language throughout the manuscript.

Author Response

We reviewed the entire manuscript in order to shorten the sentences and improve the language.
In the current version, the vast majority of the sentences are 30 words or less. Of the 34 sentences that remained with more than 30 words, 21 have between 31 and 40 words; 17 comprise lists of items (model components, research steps, examples) and 5 are definitions or citations.